# Molecular Descriptors Property Prediction Using Transformer-Based Approach

**DOI:** 10.3390/ijms241511948

**Published:** 2023-07-26

**Authors:** Tuan Tran, Chinwe Ekenna

**Affiliations:** Department of Computer Science, University at Albany, Albany, NY 12203, USA; ttran3@albany.edu

**Keywords:** machine learning, Plasmodium falciparum, molecular property, transformers, large-scale training

## Abstract

In this study, we introduce semi-supervised machine learning models designed to predict molecular properties. Our model employs a two-stage approach, involving pre-training and fine-tuning. Particularly, our model leverages a substantial amount of labeled and unlabeled data consisting of SMILES strings, a text representation system for molecules. During the pre-training stage, our model capitalizes on the Masked Language Model, which is widely used in natural language processing, for learning molecular chemical space representations. During the fine-tuning stage, our model is trained on a smaller labeled dataset to tackle specific downstream tasks, such as classification or regression. Preliminary results indicate that our model demonstrates comparable performance to state-of-the-art models on the chosen downstream tasks from MoleculeNet. Additionally, to reduce the computational overhead, we propose a new approach taking advantage of 3D compound structures for calculating the attention score used in the end-to-end transformer model to predict anti-malaria drug candidates. The results show that using the proposed attention score, our end-to-end model is able to have comparable performance with pre-trained models.

## 1. Introduction

Accurate prediction of molecular properties plays a crucial role in the chemical and pharmaceutical industries, enabling effective drug discovery and development. In the United States, an estimated 85% of drug candidates fail during clinical trials despite the extensive and expensive development process [1]. Many of these failures could be prevented by accurately predicting key properties of molecules, such as toxicity or bio-activity, whereas there have been various approaches like quantitative structure–activity relationship and high-throughput screening for drug discovery, these methods are both computationally intensive and time-consuming [2]. Consequently, there is a pressing need for prediction approaches that can swiftly and accurately evaluate molecular properties.

Machine learning models have emerged as a highly promising and potentially groundbreaking approach in pharmaceutical scientific research, providing data-driven predictions [3,4,5,6,7]. In recent research utilizing large language models, the canonical SMILES notation [8] has become the preferred format for representing molecules, as it provides a unique string representation for each molecule. Machine learning models achieve a better understanding of specific molecular properties as they are exposed to a larger number of data during training.

Hence, the success of these machine learning methods is heavily dependent on the availability of extensive labeled training data. However, acquiring such a vast amount of molecular properties through screening experiments is prohibitively expensive [9].

In this study, we present a novel variant of the language model that utilizes SMILES strings as input. Our primary focus is on developing machine learning models for predicting molecular properties. To achieve this, we propose a two-stage approach consisting of pre-training and fine-tuning, which effectively leverages a large amount of labeled and unlabeled data. During the pre-training phase, our model utilizes the Masked Language Model, a well-established unsupervised learning mechanism, on a large-scale unlabeled dataset. Subsequently, the model undergoes fine-tuning using a labeled dataset to enhance prediction performance. Through the evaluation of common downstream tasks from MoleculeNet, we demonstrate that our model exhibits comparable performance to other state-of-the-art prediction models. The overview of our model is shown in Figure 1.

Additionally, we also present the end-to-end model to predict a label (active, i.e., successfully reacts against Plasmodium falciparum parasite species, or inactive, i.e., no reaction against Plasmodium falciparum parasite species) for a given experimentally verified anti-malaria drug candidate. Malaria is a disease caused by a single-celled blood pathogen of the genus Plasmodium. Among the five known parasite species that can cause malaria in humans, Plasmodium falciparum (PF) poses the greatest threat and is responsible for the majority of malaria-related deaths [10]. The Plasmodium parasite exhibits a high mutational capacity and undergoes metabolic changes, making the development of effective drug treatments an ongoing and unresolved challenge. Furthermore, PF has the ability to evolve and develop resistance to identified drug compounds. The Centers for Disease Control and Prevention (CDC) has warned that all classes of antimalarial drugs could lose their clinical effectiveness, posing a significant risk to malaria eradication efforts [11]. In addition, Plasmodium falciparum is characterized by an abundance of proteins containing long glutamine or asparagine repeats. These regions, known as low complexity regions, have a tendency to form insoluble intracellular aggregates [12,13]. The presence of these protein deposits within cells is generally associated with cellular stress and toxicity [14]. This unique phenotype of malaria parasites holds potential for the discovery of new therapeutic strategies by exploring the properties associated with these protein aggregates. Therefore, instead of relying on only sequences representing compounds used in language models, we include the 3D structures information via contact map to introduce the global context for those compounds during the training. Overall, the contribution of our work is the following:We introduce a two-stage (pre-training and fine-tuning) model to utilize a large amount of both labeled and unlabeled data for molecular properties prediction.We propose a new approach to calculate the attention score for the transformer layer taking advantage of 3D structure information of the compounds.We show that without relying on the computational expense of a pre-trained and fine-tuned model using the proposed attention score, our model is able to achieve comparable performance in predicting anti-malaria drug candidates.

### Related Work

Methods for predicting molecular properties can be classified into multiple groups, with two prominent categories based on the type of molecular input: molecular graph and molecular string representations. In a molecular graph, each molecule is depicted as a graph comprising atom nodes interconnected by bond edges. Numerous studies have employed graph neural networks (GNNs) to acquire molecular representations [15,16]. Recent advancements in this field have introduced more sophisticated techniques for gathering neighbor data [17]. Furthermore, the integration of attention mechanisms has been proposed to enhance graph neural networks in [18]. In the study conducted by [19], several novel pre-training strategies were introduced to train GNNs at both the node and graph levels, enabling the simultaneous learning of local and global molecular representations.

There are multiple approaches for molecular string representation such as InChI [20], SELFIES [21], and SMILES [8]. InChI is a textual representation of the chemical structure of a molecule, designed to be unique and standardized across different databases and systems, and represents the connectivity of atoms in the molecule, including information about bond types and stereochemistry. SELFIES (SELF-referencing Embedded Strings) is a molecular notation system that represents organic molecules using a simplified and human-readable string format by using a recursive syntax that allows the representation to refer to itself. SMILES utilizes a sequence of characters to represent molecules in a simple way. This sequence encompasses atom symbols and bond symbols with a limited set of grammar rules. For example, melatonin, with structure C13H16N2O2, will be represented as CC(=O)NCCC1=CNC2=C1C=C(C=C2)OC in SMILES representation. The use of SMILES, which comprises a series of characters, allows for the application of state-of-the-art models in natural language modeling to extract high-quality features and enable accurate predictions for task-specific purposes. Therefore, researchers have adopted language models to learn their representations and properties [5,6]. Numerous works have been explored to encourage learning of high-quality representations with language models including input reconstruction [22], whereby a model learns to predict masked tokens; input translation [23], where the goal is to translate the input to another modality or representation; and molecules generation [24], where the model generates drug-like molecules. Nevertheless, the SMILES system has its limitations such that the vanilla SMILES system does not provide a bijective mapping between a SMILES sequence and a molecule, as different valid sequences may represent the same molecule based on the traversal path of the molecular graph. In order to overcome this challenge, several canonicalization algorithms have been developed to ensure the uniqueness of each molecular structure representation [25,26]. In this work, we focus on SMILES because most of our datasets are taken from MoleculeNet, which is a benchmark specially designed for testing machine learning methods of molecular properties, and all SMILES representations are canonical to addressing the issue of representation ambiguity.

Recently, transformer models [27,28] have gained significant popularity as architectures for acquiring self-supervised representations of molecules from text-based representations. For instance, MolBERT [29] explores various pre-training objectives on a dataset comprising 1.6 million compounds, achieving state-of-the-art performance on well-established benchmarks for virtual screening and quantitative structure–activity relationships. SMILES-BERT [9] leverages unsupervised pre-training on a dataset of 18.7 million compounds from Zinc, demonstrating the effectiveness of this approach and the excellent generalization capabilities of the pre-trained model. Another example is ChemBERTa-2 [30], which introduces a BERT-like transformer model that learns molecular fingerprints through semi-supervised pre-training, utilizing a dataset encompassing 77 million compounds.

Traditionally, computational approaches were used for drug discovery for malaria, such as quantitative structure–activity relationship (QSAR) modeling. Early statistical approaches used in QSAR modeling were linear regression models [31], Bayesian neural networks [32], and random forests [33]. However, with the availability of large chemical compound datasets, those statistical methods become more computationally expensive and not as effective [34]. To address those challenges, deep learning methods, especially neural networks, have been proposed as a practical solution. Deep learning is particularly well-suited for QSAR modeling because it is able to compute adaptive non-linear features that capture complex data patterns in complex chemical data [35]. In [36], they developed a deep learning protocol to build binary and continuous QSAR models based on large datasets and applied them to predict the anti-plasmodial activity and cytotoxicity of untested compounds for malaria drug candidates.

#### Machine Learning in Malaria Drug Discovery

The work in [37] developed a machine learning approach to predict novel synergistic drug interactions using only prior experimental combination screening data and knowledge of compound molecular structures. Particularly, using each compound’s SMILES representation, the array of features is generated based on a structural fingerprint descriptor which is calculated using 2048-bit Morgan fingerprints, a Target Fingerprint descriptor which is the probability of binding below the training cut-off for each compound vs. 1080 human protein targets, and the combination of both types of descriptor. Finally, machine learning models, such as support vector machine classifiers, use these fingerprints to make inferences between a particular representation and the experimentally observed synergy. In [38], it is shown that recurrent neural networks can generate molecule structures for virtual screening campaigns, and with fine-tuning the model can directly produce novel molecules that are active toward a given biological target. The model uses long short-term memory trained a large dataset containing SMILES to generate reasonable molecules, then fine-tuned a smaller dataset to generate biologically active molecules which can be potential drug targets for anti-malaria. Arshadi et al. [39] introduced DeepMalaria to predict the anti-Plasmodium falciparum inhibitory properties of compounds using their SMILES. Particularly, they present the application of graph convolutional neural networks for non-targeted ligand-based virtual screening for antimalarial drug discovery. In order to convert molecules to graphs, they use different features to describe each atom, such as the type of atom, atom degree, implicit valence, and chirality, used to prevent special information loss. The graph-based model is trained on publicly available antiplasmodial hit compounds and transfer learning from a large dataset was leveraged to improve the performance of the model. Moreover, the work in [40] developed five machine learning models to predict antimalarial bioactivities of a drug against Plasmodium falciparum from the values of the molecular descriptors obtained from SMILES of compounds. They implemented artificial neural networks, support vector machine, random forest, extreme gradient boost, and logistic regression and tested those models on a verified experimental anti-malaria drug compounds dataset. Lima et al. [41] used shape-based and machine learning methods for modeling antimalarial compounds to virtually screen a large database of drug-like molecules to identify promising hits for falciparum strains.

## 2. Results and Discussion

In order to evaluate our performance, we compare our method with Message Passing Neural Network (D-MPNN) [42], random forest (RF) [43], Graph Convolutional Networks (GCN) [44], and Chemberta-2 [30]. For Chemberta-2, we compared six of their proposed pre-trained models for two different tasks: Masked Language Modeling (MLM) and Multi-task Regression (MTR) with three different datasets: 5M, 10M, and 77M. All the results for reference methods are taken from [30].

For the AM dataset, we compare our method with the best model Extreme Gradient Boosting (XGB) and Artificial Neural Network (ANN) in [40], and their results are taken directly from this work [40].

### 2.1. Classification Problem

The classification performance of our model, as compared to existing methods, is presented in Table 1. We observe that our method outperforms the majority of comparable approaches on three out of four datasets. Notably, our model demonstrates exceptional performance on the ClinTox dataset, achieving the best results. Additionally, in comparison to Chemberta-2, despite being pre-trained on a smaller dataset of only 5 million, our model surpasses it on the Bace dataset and exhibits comparable performance on the other two datasets, which were pre-trained with larger datasets of 10 million and 77 million.

### 2.2. Regression Problem

The regression performance of our model, in comparison to existing methods, is depicted in Table 2. Across the board, our model consistently demonstrates excellent results on the majority of datasets. However, it is worth noting that the Clearance dataset, which consists of a smaller sample size of approximately 800 compounds, exhibits slightly less favorable performance compared to the other datasets. Nonetheless, our model archives a good performance across the majority of the regression tasks.

Table 3 presents the standard deviation values for classification and regression tasks on various datasets. We observe that the standard deviations for both classification and regression tasks are relatively small. This indicates that our model exhibits consistency and stability in its performance across multiple trials on different datasets. Additionally, the small standard deviation values suggest that the model’s predictions are robust and reliable.

We find that the degree to which improving performance on the pre-training tasks transfers to downstream tasks varies by dataset. Considering that we only pre-train the model for a limited number of epochs (specifically, 10), we have a maximum of 10 data points available. However, it is important to note that at the initial stages of training, the model may not have gained sufficient knowledge about the dataset. As a result, it is preferable to take into account the latter part of the training process, specifically the last 4 epochs, in order to effectively evaluate the impact of transfer learning. In Figure 2 and Figure 3, we show two examples of transfer learning from the pre-training stage to the fine-tuning stage with varying degrees of success. Although the improvements in pre-training loss may not result in perfectly linear enhancements in Clintox classification performance, the relationship between the two exhibits a close resemblance to a linear trend, which is highly promising for our purposes. However, for Bace regression, this trend does not hold. These findings indicate that the performance of the model is influenced by the characteristics of the datasets used.

In summary, our model demonstrates excellent performance across all fine-tuned datasets. This highlights the capability of our model to leverage the unsupervised information acquired during the pre-training step with MLM, resulting in a good performer for a wide range of tasks. The integration of large-scale unsupervised pre-training, coupled with the MLM approach, enables efficient fine-tuning of our model on labeled datasets.

#### Case Study: Anti-Malaria Drug Target Classification

Table 4 presents a comparison of our model with and without pre-training, as well as existing architectures, on the anti-malaria dataset. Our proposed models demonstrated superior performance in terms of both accuracy and F1 score when compared to XGB and ANN methods. These results are consistent with expectations, as the application of transformer-based models has been demonstrated in effectively detecting underlying patterns and substantially enhancing classification performance. The attention mechanism of the transformer-based architecture allows for the capture and utilization of intricate patterns and dependencies within the data, leading to improved performance in classification tasks.

Moreover, even without leveraging pre-training and with limited computational resources, our end-to-end model exhibited comparable performance to pre-trained models. The incorporation of 3D inter-atomic contact information enhances the model’s capability to dynamically allocate its attention to the relevant of the input data, thus reducing the reliance on prior knowledge acquired through pre-training. These findings underscore the potential of our approach to deliver robust outcomes while mitigating the computational overhead typically associated with pre-training. By streamlining the training process, our approach is able to achieve comparable performance in a more computationally efficient manner, making it a viable option for various practical applications.

## 3. Methods and Materials

### 3.1. Methodology

In this section, we will examine our proposed model. We begin by outlining the details of the building block of our model, the Transformer Encoder. Subsequently, we present the approach employed for pre-training our model using a vast amount of unlabeled data. Furthermore, we detail the fine-tuning process on a smaller dataset for classification and regression tasks related to molecular properties. As our model treats molecules as sequences, the input consists of tokenized SMILES representations of the molecules.

#### 3.1.1. Transformer Layers

Traditional transformer follows the encoder–decoder structure [45]. The encoder maps the input sequence $x=(x_{1},\ldots ,x_{n})$ to a latent sequence of continuous representations $z=(z_{1},\ldots ,z_{n})$. Then, using the *z*, the decoder generates output sequence $y=(y_{1},\ldots ,y_{n}$) as similar to *x* as possible. At each step, the model is auto-regressively consuming the previously generated symbols as additional input when generating the next [46]. Our model is based on the RoBERTa [47] transformer implementation. In contrast to the original transformer architecture, which incorporates both an encoder and decoder [27], our model only consists of a transformer layer. This layer comprises three key components: a pre-attention feed-forward neural network, a self-attention layer, and a post-attention feed-forward neural network. The pre-attention and post-attention feed-forward layers are fully connected and shared across all input and output tokens. They map the input features or the embedded features to the output in another nonlinear space and vice versa.

Since the Transformer Encoder only uses a feed-forward network, it does not inherently capture the temporal information within a sequence. To address this limitation, the self-attention layer is essential to introduce the temporal relation into consideration for feature learning. The self-attention mechanism plays an important role in capturing the relationships among the various elements within the input, whether they are words in a sentence or characters in a string. By including the self-attention layer, the model becomes capable of capturing the contextual dependencies and understanding the sequential structure of the input data. The self-attention mechanism partitions the input data into three matrices: the query matrix *Q*, the key matrix *K*, and the value matrix *V*. The query matrix and the key matrix combine together to form the input for the Softmax function. By applying the Softmax function, attention weights are generated. These attention weights are then applied to the value matrix, resulting in the generation of output features that are attentive to the entire input sequence. Thus, the attention is calculated as the following:(1)$$Attention\left(K,Q,V\right)=softmax\left(\frac{QK^{T}}{\sqrt{d_{k}}}\right)V$$
where $d_{k}$ is the dimension of the query and key matrix.

In our model, we incorporate multi-head attention instead of using a single self-attention layer. This multi-head attention mechanism enables the model to extract information from different representation subspaces, which would not be possible with a single attention head.

#### 3.1.2. Pre-Training Setup

We adopt the Masked Language Model (MLM) pre-training procedure from RoBERTa. In MLM, given a partial sentence with masked tokens, using other visible tokens, the model predicts those masked ones. RoBERTa, being an unsupervised learning model, is capable of utilizing vast amounts of unlabeled sentences from natural languages for training. In our approach, we follow a specific method for masking SMILES inputs. We randomly select 15% of the tokens in SMILES for masking, ensuring that each SMILES has at least one token masked. For each selected token, there is an 80% chance it will be changed to a masked token, a 10% chance it will be randomly replaced with another token from the dictionary, and a corresponding 10% chance it will be kept unchanged. During the training of our model, the original SMILES sequences are used as the ground truth. However, the loss function is computed based solely on the output of the masked tokens within the sequences. This approach allows the model to focus on learning the representations and relationships specifically related to the masked tokens, without being influenced by the remaining tokens in the sequence. The randomness in the masking procedure could increase the generalization ability of the model and keep it from over-fitting [9].

Furthermore, through the process of learning to retrieve masked tokens, the model develops a representational topology of the chemical space that exhibits generalizability in property prediction tasks [48]. The tokens are initially embedded into the feature space using the tokenizer. Alongside token embedding, positional embedding is also incorporated to incorporate sequential information. This positional embedding allows the model to effectively utilize the temporal information present in the input sequences to be used in the self-attention layer. Figure 4 shows the overview of our pre-training stage.

#### 3.1.3. Fine-Tuned Model

Fine-tuning is adequate due to the inherent flexibility provided by the self-attention mechanism within the transformer layer. This enables our model to be utilized for various downstream tasks by simply adjusting the input and output components accordingly. In particular, a straightforward approach involves appending a linear classifier or regressor to the base model and training them together on a smaller labeled dataset. This joint training process allows the model to adapt and specialize its predictions for the specific task at hand. Figure 5 shows the overview of our fine-tuning stage.

#### 3.1.4. Case Study: Anti-Malaria Drug Target Classification

Since we also want to take advantage of the 3D structures of the drug compound, in addition to SMILES, we also include the 3D structure of the compound for our proposed transformer model. Specifically, from the 3D structure, we extract the contact map of each atom. In simple terms, the contact map of the compound is similar to the adjacency matrix of the graph. Particularly, the contact map describes the pairs of atoms that are in contact (within 8 Å of one another) in the compound structure but lie apart (by at least six positions) in the underlying sequence [49]. One example of the contact map is shown in Figure 6.

Since robustness to global changes such as 3D translations and rotations is an underlying principle for molecular representation learning, we seek to satisfy rotation and translation invariance. So, we take inspiration from SE(3)-Transformer [50] and AlphaFold2 [51], and apply a convolutional operation to the contact map matrix *C* as C′=Conv2d(C) where Conv2d denotes a 2D shallow convolutional network with a kernel size of 1 × 1. So, the attention score that input token *i* pays to input token *j* is computed as follows: (2)$$\begin{array}{cc} q_{i} & =f_{Q}\left(x_{i}\right),k_{i}=f_{K}\left(x_{i}\right),v_{i}=f_{V}\left(x_{i}\right) \end{array}$$(3)$$\begin{array}{cc} a_{ij} & =\frac{q_{i}*k_{j}^{T}}{\sqrt{d_{k}}}*d_{ij}^{\prime } \end{array}$$(4)$$\begin{array}{cc} z_{i} & =\sum_{j=1}^{N}\sigma \left(a_{ij}\right)v_{j} \end{array}$$
where $\left\{f_{Q},f_{K},f_{V}\right\}$ are embedding transformations; $\left\{q_{i},k_{i},v_{i}\right\}$ are, respectively, the query, key, and value vector with the same dimension $d_{k}$; σ denotes the Softmax function; $d_{ij}$${}^{\prime }$$\in C^{\prime }$ controls the impact of inter-atomic contact over the attention score, and $z_{i}$ is the output embedding of the token i. Figure 7 shows the overview of our end-to-end classification model for anti-malaria drugs. We use the same transformer layer mentioned previously but instead of the default attention function, we use the function in Equation (Equation 4).

### 3.2. Evaluation Dataset

#### 3.2.1. Pre-Trained Dataset

For our pre-training dataset, we utilize a subset of SMILES data from PubChem [52], which is the largest open-source collection of chemical structures. To assist the large-scale pre-training, the SMILES sequences are first canonicalized and shuffled. Specifically, we select a subset dataset of 5 million SMILES from Chemberta [48]. During the unsupervised pre-training stage, the SMILES sequences are tokenized into individual tokens using our tokenizer, serving as the inputs for our model.

#### 3.2.2. Fine-Tuned Dataset

We assess the performance of our models across multiple regression and classification tasks from MoleculeNet, a comprehensive dataset collection for molecular properties evaluation [53]. These datasets encompass a diverse range of sample sizes, varying from 1000 to 8000 examples, and cover various medicinal chemistry applications such as brain penetrability, toxicity, solubility, and on-target inhibition. These datasets serve as benchmarks for evaluating the efficacy of state-of-the-art machine learning approaches in predicting molecular properties, which are crucial in the drug discovery process. Additionally, we incorporate an antimalarial drug candidate dataset [40] into our evaluation. We utilize the following datasets to further assess the performance of our models for classification and regression tasks.

**BACE** (Classification and Regression): The BACE dataset provides quantitative IC50 and qualitative binding results for a set of inhibitors of human beta-secretase 1 (BACE-1). It has 1513 compounds.**Clearance** (Regression): The dataset contains human clearance, which is the parameter that determines total systemic exposure to the drug. It has 837 compounds.**Delaney** (Regression): The Delaney dataset contains structures and water solubility data. It has 1128 compounds.**Lipophilicity** (Regression): The lipophilicity dataset provides experimental results of the octanol/water distribution coefficient. It has 4200 compounds.**BBBP** (Classification): The blood–brain barrier penetration (BBBP) dataset consists of binary labels for the prediction of barrier permeability. It has 2039 compounds.**ClinTox** (Classification): The ClinTox dataset compares drugs approved by the FDA and drugs that have failed clinical trials for toxicity reasons. It has 1478 compounds.**Tox21** (Classification): The “Toxicology in the 21st Century” (Tox21) contains qualitative toxicity measurements on 12 biological targets, including nuclear receptors and stress response pathways. It has 7831 compounds.**Antimalarial** (Classification): The antimalarial dataset is a given experimentally verified antimalarial drug candidate from public chemical databases, ChEMBL and PubChem. It has 4794 compounds.

For datasets with multiple tasks, we selected a single representative task: the clinical toxicity (CT_TOX) task from ClinTox and the p53 stress-response pathway activation (SR-p53) task from Tox21. For each dataset, we randomly select 80% for training, 10% for validation, and 10% for evaluation.

### 3.3. Implementation

#### 3.3.1. Model Details

The implementation of our model uses 12 attention heads with 6 layers, for a total of 72 attention mechanisms, and 3072 fully connected embedding sizes. Our model is implemented using transformer libraries provided by HuggingFace [54]. We trained the network by minimizing the root mean square loss for the masked values using the Adam optimizer [55] with $\beta_{1}=0.9$, $\beta_{2}=0.98$, and ϵ = 1 × 10${}^{-9}$. The model was trained for 10 epochs with a batch size of 512.

In all of our experimental settings, we fine-tune our model with each labeled dataset for a total of 50 epochs while utilizing early stopping based on the validation loss. The best-performing model, as determined by the validation data, is selected for the final evaluation. During the fine-tuning process, we normalize the labels to have a zero mean and a standard deviation of one for regression tasks. For classification tasks, we employ balanced class weights to account for potential class imbalances. To ensure reliable and robust evaluation, we conduct each experiment five times for each dataset. The reported performance metrics are the average values of the receiver operating characteristic curve (ROC-AUC) for classification tasks and the root-mean-square error (RMSE) for regression tasks. In classification tasks, a higher ROC-AUC value indicates better model performance, whereas, in regression tasks, a lower RMSE value signifies better model performance.

#### 3.3.2. Tokenizer

We utilized the tokenizer libraries provided by HuggingFace for training our tokenizers. Specifically, for a given tokenizer, an initial split of the input sequence is performed by a pre-tokenizer. In addition to the WordPiece tokenizer provided by Hugging Face, we also incorporated a splitting based on digits. To ensure efficient and effective tokenization, we set a maximum vocabulary size of 600 tokens, which is based on a dictionary containing commonly used SMILES characters. Furthermore, a maximum sequence length of 512 tokens was employed to accommodate the input sequences.

#### 3.3.3. Case Study: Anti-Malaria Drug Target Classification

We use the same parameters as above, except our model uses eight attention heads with three layers. The model was trained for 300 epochs with a batch size of 128. Since this is a classification task, we use accuracy, F1, and ROC-AUC as our performance metrics.

## 4. Conclusions

In the paper, we introduce a semi-supervised learning approach for predicting molecular properties, aiming to leverage the abundance of unlabeled molecular data effectively. The core component of our model is RoBERTa, which combines the transformer layer and attention mechanism. Our semi-supervised method makes use of a substantial amount of unlabeled data by pre-training the model with MLM. Subsequently, the pre-trained model can be easily fine-tuned on the labeled dataset to enhance prediction performance. Preliminary findings indicate that our model achieves a comparable level of performance to state-of-the-art models on the specific downstream tasks selected from MoleculeNet.

Moreover, we also conducted a case study for our model using an anti-malaria drug target dataset. To reduce the computational overhead, we also proposed an end-to-end transformer-based model for drug target discovery. Particularly, we proposed a new approach to calculate the attention score by taking advantage of both SMILES and 3D structures of potential drug targets in the transformer layer. The experimental results show that without pre-trained using large datasets, our model achieved comparable performance using the new attention score.

Our current analysis covers only a small portion of the experiments we plan to conduct. We plan to delve into systematic hyperparameter tuning, multitask fine-tuning, and the utilization of larger pre-training datasets. Additionally, we also would like to try the application of larger models, as their capacity to capture more intricate semantic information has been demonstrated in numerous studies within the field of natural language processing. Through our work, we believe that we can contribute to the discovery and development of anti-malaria drugs, aiding in the advancement of this critical area of research.

## Figures and Tables

**Figure 1 ijms-24-11948-f001:**
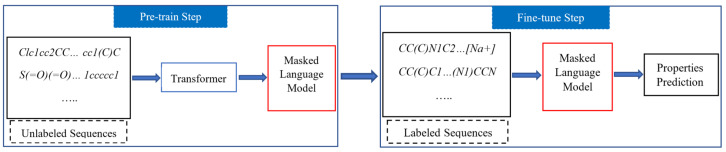
Overview of our model with two stages: pre-training on large-scale unlabeled datasets and fine-tuning on smaller labeled datasets for downstream tasks.

**Figure 2 ijms-24-11948-f002:**
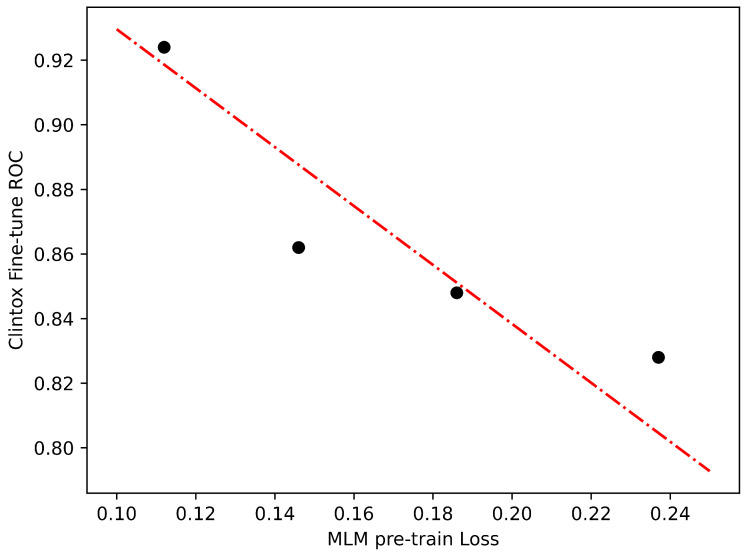
Transfer learning for fine-tuning performance versus pre-training loss from MLM to ClinTox classification. The dotted lines represent linear models fit to the data points.

**Figure 3 ijms-24-11948-f003:**
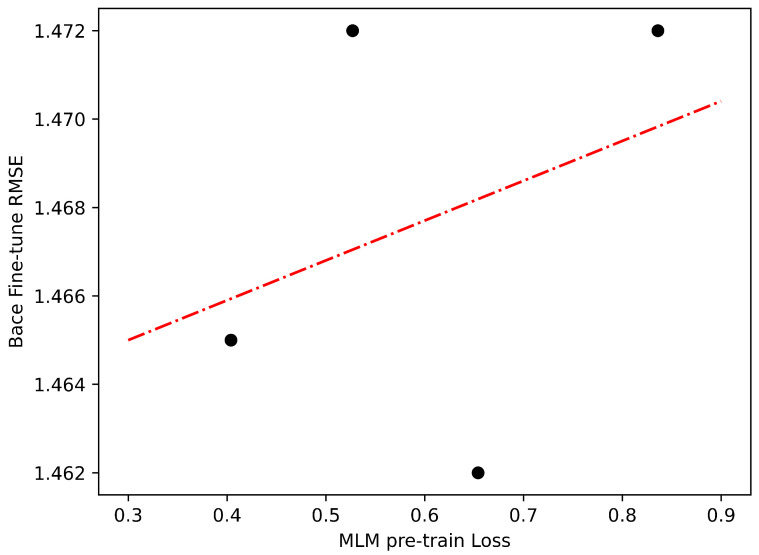
Transfer learning for fine-tuning performance versus pre-training loss from MLM to Bace regression. The dotted lines represent linear models fit to the data points.

**Figure 4 ijms-24-11948-f004:**
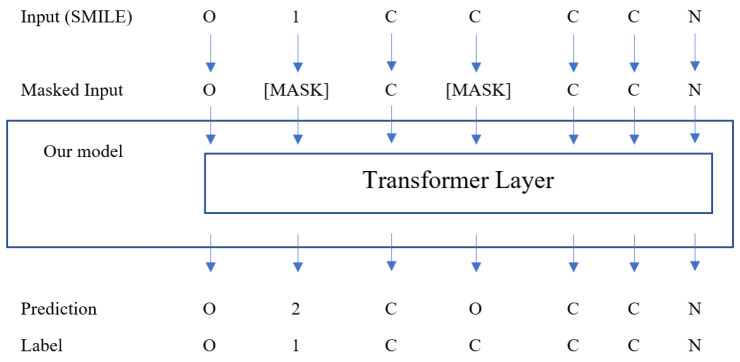
Overview of the pre-training stage with MLM.

**Figure 5 ijms-24-11948-f005:**
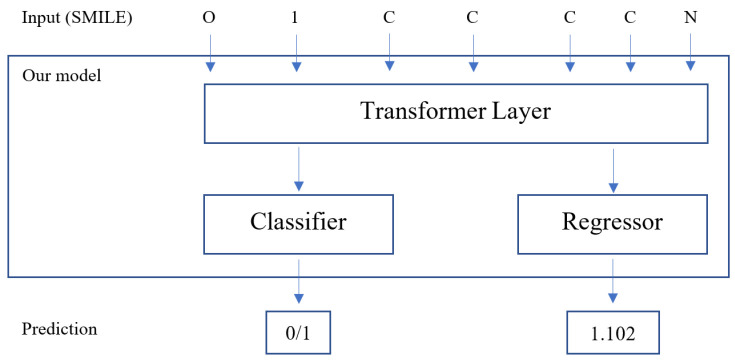
Overview of the fine-tuning stage with MLM for classification and regression tasks.

**Figure 6 ijms-24-11948-f006:**
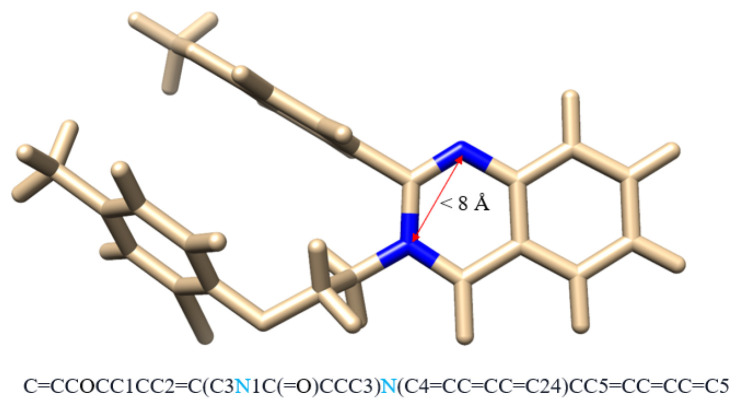
Example of a contact map. Although two Nitrogen atoms are at least six positions apart in SMILES string, they are within 8 Å of each other.

**Figure 7 ijms-24-11948-f007:**
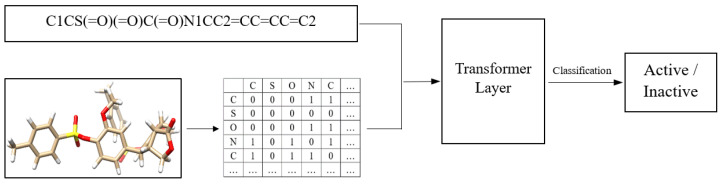
Overview of our model for anti-malaria drugs classification.

**Table 1 ijms-24-11948-t001:** Comparison of our pre-trained model on different dataset vs. existing architectures on selected MoleculeNet classification tasks. The best results are indicated by the bold numbers.

	BACE	BBBP	ClinTox	Tox21
D-MPNN	0.812	0.697	0.906	0.719
RF	**0.851**	0.719	0.783	0.724
GCN	0.818	0.676	0.907	0.688
MLM-5M	0.793	0.701	0.341	0.762
MLM-10M	0.729	0.696	0.349	0.748
MLM-77M	0.735	0.698	0.239	0.749
MTR-5M	0.734	**0.742**	0.552	**0.834**
MTR-10M	0.783	0.733	0.601	0.827
MTR-77M	0.799	0.728	0.563	0.817
Our model	0.808	0.683	**0.914**	0.781

**Table 2 ijms-24-11948-t002:** Comparison of our pre-trained model on different dataset vs. existing architectures on selected MoleculeNet regression tasks. The best results are indicated by the bold numbers.

	BACE	Clearance	Delaney	Lipophilicity
D-MPNN	2.253	49.754	1.105	1.212
RF	**1.318**	52.077	1.741	0.962
GCN	1.645	51.227	0.885	0.781
MLM-5M	1.451	54.601	0.946	0.986
MLM-10M	1.611	53.859	0.961	1.009
MLM-77M	1.509	52.754	1.025	0.987
MTR-5M	1.477	50.154	0.874	0.758
MTR-10M	1.417	48.934	**0.858**	**0.744**
MTR-77M	1.363	**48.515**	0.889	0.798
Our model	1.481	56.063	1.066	0.908

**Table 3 ijms-24-11948-t003:** The standard deviation of our model on selected classification and regression tasks.

	BACE	BBBP	ClinTox	Tox21	Clearance	Delaney	Lipophilicity
Classification	0.033	0.025	0.013	0.026	N/A	N/A	N/A
Regression	0.173	N/A	N/A	N/A	1.341	0.125	0.037

**Table 4 ijms-24-11948-t004:** Comparison of our model with and without pre-training vs. existing architectures on anti-malaria dataset.

	Acc	F1	AUC
XGB	0.8318	0.8412	N/A
ANN	0.8223	0.8445	N/A
With_pre-training	0.8601	0.8721	0.9012
Without_pre-training	0.8553	0.8471	0.8937

## Data Availability

The pre-train dataset is taken from Chemberta [48]. The fine-tuned datasets are taken from MoleculeNet [53]. The antimalaria drug candidate dataset is taken from [40].

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
