# Peer review of "Molecular Descriptors Property Prediction Using Transformer-Based Approach"

_ijms, 2023, doi:10.3390/ijms241511948_

Round 1

Reviewer 1 Report

The authors developed a ML model for molecular property prediction. Masked language model was included in the pre-training stage and several small datasets were used to finetune model. The model is reported to have comparable performance with low RMSD and high ROC-AUC values. 

Major:

- I don't quite get the relation between Malaria and molecular property. The first sentence in Abstract and first paragraph in Introduction seems not directly related to the main context. It's not that necessary to include Malaria in Abstract and introduction, at least the first paragraph, for anti-Malaria as a case study, 

- Can the authors provide more data point in Figure 6 and 7? Only 4 in each fig seems not robust. Also, why not combine these two as subfigures into one.

- Can the authors report STD besides mean value in Table 1 and 2? These would help to check model stability.

Minor:

- Please keep consistent significant figures, such as line 246.

- The authors should consider open source their codes. 

Reviewer 2 Report

In this study, the authors propose a novel semi-supervised ML method for the prediction of the molecular properties of drugs. 

In order to test, the predictive ability of the developed model, they applied their method to anti-malaria drug target classification. 

This is an interesting, well-performed study with a nice idea, and the results justify the validity of their approach. 

Comments:

1. Lines 327-331: It is stated that for the MLM there is a linear improvement in transfer learning, while this does not hold for the Bace regression. However, the linear trend is quite questionable even in the case of the MLM, since a biphasic behavior is observed with an initial steep linear reduction that leads to a much less steep loss .

2. Why was the softmax activation function used? Do other functions, such as ReLu, have been tested and found to be inferior to softmax?

Reviewer 3 Report

Tuan et al. are reporting a SMILES-based transformer model for molecular properties predictions.

1. In line 50, authors criticized that experimentally achieving molecular properties requires experimental validations, which are expensive. However, the experimental assessment is inevitable to ultimately validate whether a compound is active or not. In addition, in the fine-tuning step, authors require tested molecules to be labeled for training. The reviewer doesn’t see how authors’ method can bypass experimental validations.

2. In line 78, authors claimed that there are two molecular property prediction categories - molecular graph and SMILES. How about fingerprints? Fingerprints are widely involved in machine learning campaigns to describe molecules for model training. 

3. In line 88, SMILES was introduced by authors. What are differences between SMILES and canonical SMILES? What are differences between SMILES and other string-based representative like INCHI, etc.? The discussion can be extended. 

4. In Figure 7, the dashed red line can hardly be considered as a linear model to fit data points. R square can be reported for both Figure 6 and 7.  The R square for figure 7 can possibly be undesired, which can indicate a compromised transfer learning process. 

5. The case study was carried out for anti-malaria drug target classification. Are there any new chemical matters been predicted to potentially have anti-malaria activities? The experimental validation can be inevitable to validate the prediction outcome for maybe one or two newly predicted compounds to support the effectiveness of the proposed model.

6. Anti-malaria drugs can exhibit the functional activity through different mechanisms by targeting different targets. Is there a specific signaling pathway or target groups that authors are particularly addressing?

Overall, the reviewer would suggest a major revision.

It has been indicated by the question above. 

Reviewer 4 Report

The manuscrpt "Molecular Descriptors Property Prediction using Transformer Based Approach" by Tuan Tran and Chinwe Ekenna is an interesting example of QSAR-related research on novel mathematical models of molecular description.

I find the manuscript well written, not overladen with extensive tables - they are confined to the necessary amount. It is also possible to follow the authors' reasoning relatively easily. Model generation and validation is adequately described. The models compare reasonably well to the existing algorithms, but are not decisively better, and for this reason I have lowered the overall assessment of interest to the readers and significance of content. On the other hand, the manuscript provides detailed hints for other prospective scientists trying to develop their models, and this makes the manuscript worthy of publishing.

In overall assessment, I do not find any issues to be corrected with necessity, and I can recommend its publication "as is".

Author Response

We thank the reviewers for the comments.

Round 2

Reviewer 1 Report

The authors have addressed my issues. 

Reviewer 3 Report

According to authors's response, authors have no intention to continue the experimental validations to support the effectiveness of their model on generating active new molecules. Current manuscript only supports that known knowledge can be recurred by the model.

According to authors's response, authors are not targeting any specific signaling pathways nor targets for designing anti-malaria drugs. It is skeptical to consider a drug discovery campaign without biological MOA.

The reviewer would suggest the acceptance but the reviewer's opinion is reserved. 

N/A